

# Population influences desiccation tolerance in an invasive aquatic snail, *Potamopyrgus antipodarum* (Tateidae, Mollusca)

Edward P. Levri, Sheila Hutchinson, Rebecca Luft, Colin Berkheimer and Kellie Wilson

Division of Mathematics and Natural Sciences, Penn State Altoona, Altoona, Pennsylvania, United States of America

## ABSTRACT

Traits in species that influence invasion success may vary in populations across its invaded range. The aquatic New Zealand mud snail, *Potamopyrgus antipodarum*, reproduces parthenogenetically in its invaded range, and a few distinct clonal genotypes have been identified in North America. Much of the spread of the snail in North America has been the result of unintentional overland transport by recreational water users. Thus, desiccation tolerance may play an important role in the invasion success of this species. The primary goal of these experiments is to determine if variation in desiccation tolerance exists between populations of this species. Here we compared multiple multi-locus genotypes (MLGs) and populations within those genotypes with regard to their desiccation tolerance. We conducted three experiments. The first compared the survival rate over time of snails from three populations and two MLGs with regard to their ability to survive being completely removed from water. The second experiment examined different size classes of snails from the same population, and the third experiment compared four different populations and two MLGs genotypes with respect to their survival rate over time when removed from water but being kept in moist conditions. We found larger snails tolerate desiccation longer than smaller snails with snails between 4 and 4.6 mm surviving at a rate of more than 50% after 12 h while smaller snails survived at a less than 5% rate after 12 h. We also found significant variation both between and within MLGs in their survival rate when completely removed from water and dried with the MLG from the western US having a more than 50% greater survival probability than the eastern MLGs at both 18 and 24 h out of water. When removed from water and kept moist all MLGs had a near 100% survival rate at 60 days at 7 °C, and most survived at a greater than 90% rate at 60 days at 17 °C, while no MLG's survived past 30 days at 27 °C. The results demonstrate that variation for desiccation tolerance exists between populations of this invader which could influence the invasiveness of different populations.

Corresponding author
Edward P. Levri, epl1@psu.edu

## INTRODUCTION

Variation in traits may exist within and between populations of an invasive species (*Voisin, Engel & Viard, 2005*; *Forsman, 2014*). This variation may originate from the genetic variation introduced in the founding invading populations (*Lee, 2002*; *Forsman, 2014*; *Bock et al., 2015*), could be created due to mutation and other evolutionary processes once the invader arrives in its new location (*Lee, 2002*; *Bock et al., 2015*), or could be due to environmental, maternal, or epigenetic effects (*Thorson et al., 2017*). This variation, could be found in traits that influence invasion success resulting in different populations of an invasive species showing different invasive characteristics.

The New Zealand mud snail (NZMS), *Potamopyrgus antipodarum* is an incredibly successful invader (*Alonso & Castro-Díez, 2008*) endemic to New Zealand with invasive populations on six continents, Australia (*Ponder, 1988*), Europe (*Ponder, 1988*), Asia (*Shamida & Urabe, 2003*; *Son, 2008*), North America (*Bowler, 1991*; *Zaranko, Farara & Thompson, 1997*), South America (*Collado, 2014*), and Africa (*Taybi, Mabrouki & Gloer, 2021*). The snail has been demonstrated to cause substantial ecological harm (see *Geist et al., 2022* and *Alonso et al., 2023* for reviews). NZMS are gilled snails ranging in adult length in New Zealand from 4 to 11.5 mm (*Winterbourn, 1970*), but in their invaded range they typically reach a maximum length between 4 and 5 mm (*Proctor et al., 2007*). They show a wide tolerance to variation in salinity with the ability to grow at salinities approaching sea water (*Drown, Levri & Dybdahl, 2011*), and they have the ability to withstand a wide array of temperatures from near freezing to 34 °C with densities peaking at 17 °C (*Cox & Rutherford, 2000*; *Dybdahl & Kane, 2005*; *Moffitt & James, 2012*; *Bennett et al., 2015*). NZMS perform better with increasing conductivity and where calcium ions are not limiting (*Herbst, Bogan & Lusardi, 2008*; *Larson, Dewey & Krist, 2020*; *Levri et al., 2020*), and, as most molluscs, they prefer pH levels that exceed seven (*Levri et al., 2020*). The snail lives in mixed populations of sexual individuals and asexual clones in New Zealand (*Lively, 1987*), but in its invaded range it appears to be composed only of asexual individuals (*Proctor et al., 2007*). In North America, there appear to be at least three distinctly different clones based on allozyme multi-locus genotypes (MLGs) (*Dybdahl & Drown, 2011*). The US1 MLG is widespread throughout the American west (*Proctor et al., 2007*) and has recently colonized locations in the Eastern U.S. as well (*Levri et al., 2020*; M. Dybdahl and J. Finger, 2017, unpublished data). The US2 MLG is found in the Laurentian Great Lakes and some streams that empty into the lakes (*Levri & Jacoby, 2008*; *Levri et al., 2012*; M. Dybdahl and J. Finger, 2017, unpublished data). The US3 MLG is known from only one location in the Snake River in Idaho (*Proctor et al., 2007*). A more extensive genetic study of NZMS has shown some support for the above groupings using MLG's. *Donne et al. (2020)* utilizing SNP markers found that the US2 MLG is most closely related to European invasive populations, and various US1 populations cluster within certain New Zealand populations. However, *Donne et al. (2020)* suggests that there could have been multiple introductions of the US1 MLG. The existence of MLGs affords the opportunity to examine the influence of genotype on invasion success as well as to study how populations within a MLG can vary independently from each other as dispersal

occurs. Behaviors that are suspected to influence invasion success in the snail have been demonstrated to vary between MLGs and between populations within MLG in North America (*Levri & Clark, 2015*; *Levri et al., 2017*; *Levri, Luft & Li, 2019*). *Levri & Clark (2015)* showed that different genotypes varied in their response to water currents, their tendency to disperse by floating at the surface, and in their responses to light and gravity with the US1 MLG showing substantial differences from other invasive populations and New Zealand native populations. The invasive US1 and US2 MLGs also responded to the odor of a North American fish predator by moving faster while a less invasive MLG (US3) showed no response (*Levri et al., 2017*). The snail also exhibited variation in floating dispersal behaviors in response to crayfish and fish chemical cues with some variation evident between populations of the same MLG (*Levri, Luft & Li, 2019*).

Invasion success of aquatic species in freshwater systems is often greatly enhanced if they are capable of surviving air exposure for a period of time to aid dispersal between water bodies (*Facon et al., 2004*; *Koetsier & Urquhart, 2012*; *Weir & Salice, 2012*; *Havel et al., 2014*; *Ferincz et al., 2016*; *Emiljanowicz, Hager & Newman, 2017*). Desiccation tolerance appears to be a crucial component to the success of the globally distributed NZMS (*Richards, O'Connell & Shinn, 2004*; *Alonso & Castro-Díez, 2008*, *2012*). The species has spread rapidly in many areas, and this spread has been greatly aided by anthropogenic transport between watersheds (*Proctor et al., 2007*; *Alonso & Castro-Díez, 2008*). The movement between waterbodies usually requires at least some time out of the water and likely necessitates some air exposure. The snail is a caenogastropod and does not possess a lung, yet multiple studies have demonstrated that the NZMS is well adapted for air exposure and can survive for at least hours exposed to air (*Winterbourn, 1970*; *Richards, O'Connell & Shinn, 2004*; *Alonso & Castro-Díez, 2012*; *Poznanska et al., 2015*). *Winterbourn (1970)* found that the snail can survive up to 30 h exposed to air on a dry substrate and up to about 50 days if kept moist, *Richards, O'Connell & Shinn (2004)* demonstrated that increasing temperatures and decreasing snail size results in greater mortality when the snails are kept out of water, and *Alonso & Castro-Díez (2012)* found that if removed completely from water and dried, 50% of the snails died between 26 and 28 h and 99% of them died between 43 and 50 h.

Here we conducted three separate lab experiments to examined how NZMS varied in their tolerance of being removed from water as measured by survival. Two experiments assessed how different populations varied in their desiccation tolerance, and one experiment examined the effect length of individual NZMS on desiccation tolerance. To determine how population may influence desiccation tolerance in dry conditions, we compare the survival rates of three different invasive populations when exposed to air for various amounts of time. To determine the effect of size of snail on survival when exposed to air, we compare the desiccation tolerance of different size classes of the same population of snails in a similar way as *Richards, O'Connell & Shinn (2004)* with the goal of determining if we could find the same relationship in a different population of snails. Finally, to determine how population of origin may influence survival in snails in moist conditions, we compare the survival rates of different populations of snails kept out of water but in moist conditions at different temperatures. We hypothesized that there would
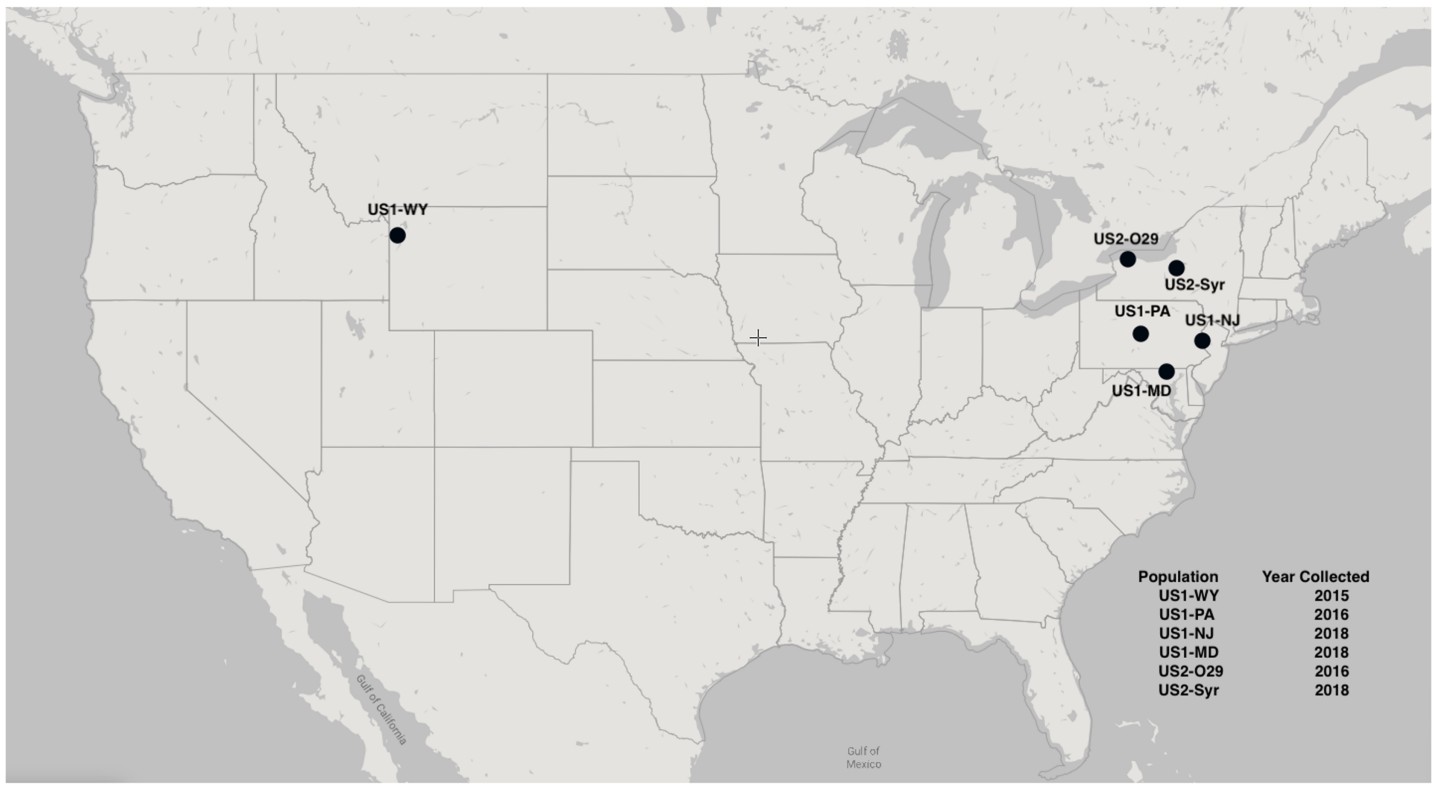

**Figure 1 A map of the locations of the NZMS populations were collected.** The map was created using Scribblemaps.com.

be differences between populations in desiccation tolerance as the Wyoming population is from the arid western US while the others are found in the wetter eastern US, and we predicted that larger snails would survive longer out of water than smaller snails as others have found (*Richards, O'Connell & Shinn, 2004*).

## MATERIALS AND METHODS

### Snail population collection and maintenance

A total of six different populations of NZMS were used in the three experiments: US1-PA (collected 2016; Spring Creek, Centre County, PA, USA), US1-WY (collected 2015; Polecat Creek, WY, USA), and US2-O29 (collected 2016; Fish Creek, Niagara County, NY, USA) were used in experiment 1, US1-PA was used in experiment 2, and US1-PA, US1-MD (collected 2018; Gunpowder Falls River, Baltimore County, MD, USA), US1-NJ (collected 2018; Musconencong River, NJ, USA), and US2-Syr (collected 2018; Geddes Brook, Syracuse, NY, USA) were used in experiment 3. All populations are invasive in North America, and a map of the locations of each population can be seen in Fig. 1. Pennsylvania samples were collected with a Scientific Collectors permit from the PA Fish and Boat Commission (Permit No. 2016-01-0317 Type 1). The North American MLGs were originally genotyped using allozyme, microsatellite DNA, and mitochondrial DNA genetic markers (*Dybdahl & Drown, 2011*). All MLGs used in the experiments were lab-reared and

had spent several generations in a lab environment before these experiments; typically, in our lab the snails will go through one to two generations per year.

Before each experiment, we maintained all populations in multiple one-liter plastic bins in water treated with Amquel® and containing 3 ppt (parts per thousand) seawater, since these snails have been found to grow faster in dilute seawater than in freshwater (*Drown, Levri & Dybdahl, 2011*). To treat the water, we added 15 ml of Amquel® to approximately 100 L of aged tap water. Following this we add sea salt using Instant Ocean® to the water and test it until a salinity of 3 ppt is reached. We fed the snails *Spirulina* powder *ad libitum* and changed their water three times per week. All snails were taken from the field and then cultured in our laboratory on the Penn State Altoona campus except for US1-WY. The US1-WY snails were orginally collected in Wyoming and brought to a lab on the University of Wyoming campus and reared there for several weeks before shipping them through the mail to Penn State Altoona. All of the populations spent multiple generations in our lab. The US1-WY snails, as well as snails from all of the other populations that were used in the experiment were born and reared in our lab. No more than about 200 snails were kept in each container. The containers were kept in a cabinet at a constant temperature of 17 °C.

## Experiment 1: the effect of population on desiccation tolerance

This experiment was conducted in July of 2017 using three invasive populations, two populations of the US1 MLG (US1-WY and US1-PA) and one US2 population (US2-O29) (we did not have US1-NJ, US1-MD, or US2-Syr snails at the time of this experiment). The snails were taken from multiple containers of each population. We exposed snails from each population to air in the lab at a relative humidity of 19–24% with a constant temperature of 18 °C for one of six time periods: 0, 9, 18, 24, 36, or 48 h. A total of 30 snails in groups of 10 between 4 and 5 mm in length of the US1-WY and US1-PA snails were exposed in each time period, and a total of 27 of the US2-O29 (in groups of nine) were exposed in each time period (the difference was due to having fewer numbers of suitable sized snails from the US2-O29 population). The snails were selected to not differ in length between population (ANOVA: F = 1.04, df = 2, $p = 0.355$) or between time of exposure (ANOVA: F = 0.09, df = 5, $p = 0.994$). To expose the snails to air, the snails were removed from water and placed on a paper towel. They were blot-dried with another paper towel and then placed into an empty 50 ml plastic cup. The cups were placed on plastic trays, that could each hold up to 20 cups, with rows of each population on each tray. Separate trays were used for each treatment, and the trays were placed on a lab bench in the center of the lab. Placing all of the snails used for each treatment on the same trays did create a confounding variable. However, we did this to minimize errors that could be made in the execution of the experiment, and the trays were arranged in a way so that all of the cups were within 1 m of each other, and no differences in temperature or humidity were found between tray positions. The snails in the 0 h treatment were then immediately returned to water. The snails from the other treatments were returned to water at the appropriate time. It was determined that the snail was alive after being returned to water if the snail was observed to move. The snails retreated into their shells when out of the water. If alive, once

reintroduced to water, the snails will eventually emerge from their shells and begin to crawl around. Mortality was assessed after 1 week.

## Experiment 2: the effect of snail size on desiccation tolerance

This experiment was conducted in the February of 2018 using only the US1-PA population because it was the most abundant population in the lab at that time. We separated snails from this population into three different size classes: 2.2–2.4 mm in length, 3.2–3.4 mm, and 4.0–4.6 mm. We exposed snails from each size class to air in the lab at a relative humidity of 19–23% with a constant temperature of 18 °C for one of eight time periods: 0, 1, 3, 6, 12, 24, 36, or 48 h. A total of 30 snails in groups of ten were exposed from each size class for each time interval. The same procedure as described in Experiment 1 was used to expose the snails to air and assess mortality. The cups with the snails were placed on trays as described in Experiment 1.

## Experiment 3: the effect of population on tolerance of moist conditions at different temperatures

This experiment was conducted in the summer of 2019 using four different invasive populations, three populations of the US1 MLG (US1-PA, US1-NJ, and US1-MD) and one US2 population (US2-Syr). These populations were used because we had more of these snails reared in the lab than other populations, and only these could give us the sample sizes that we desired. The snails used ranged in size from 4 to 5 mm in length, and the snails did not differ in length between population (ANOVA: $F = 0.538$, df = 3, $p = 0.656$) or temperature treatment (ANOVA: $F = 0.207$, df = 2, $p = 0.813$). To expose the snails to moist conditions, we removed them from water and then put 10 snails at a time on a Scott[®] brand Multifold dry paper towel. The towel was then folded into approximately a 5 cm × 5 cm square so that the snails were in the center, and there were at least three layers of towel around the group of snails. A total of 6 ml of dechlorinated tap water treated with Amquel[®] was added to each towel to make it moist. This amount was chosen because 6 ml resulted in a saturated towel without excess water. Then for each temperature treatment all of the towels were placed together in a sealed 15 cm × 15 cm × 5 cm plastic box. The box was then placed in the appropriate temperature. The boxes were placed in a refrigerator for 7 °C, in a Styrofoam cooler covered with a lid for 17 °C, or in a covered warm water bath for 27 °C. A total of 30 snails (three replicates of 10) from each population were exposed to each temperature at each time interval, resulting in a total of 210 snails of each population being exposed to each temperature, thirty at each time period. For the 7 °C treatment, snails were assessed for survival at 0, 25, 40, 45, 50, 56, and 60 days. For the 17 °C treatment survival was assessed at 0, 10, 21, 30, 40, 50, and 60 days, and for the 27 °C treatment survival was assessed at 0, 10, 16, 22, 27, 30, and 36 days after wrapping the snails. These intervals were chosen based on preliminary experiments. Snails exposed for 0 days were immediately removed from the paper towel and re-exposed to water. Mortality was assessed in the same manner as in both earlier experiments.

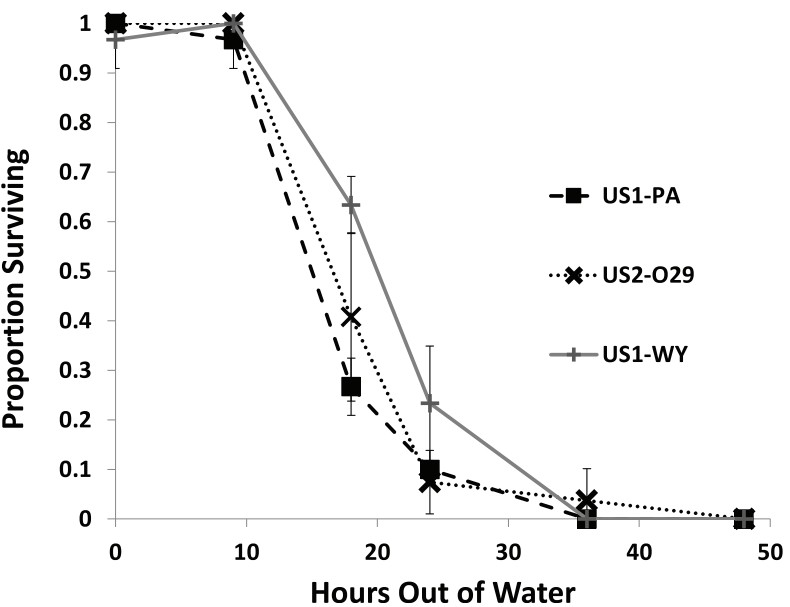

**Figure 2** The average proportion of snails surviving from three populations when exposed to air over 48 h. Error bars are standard errors of average proportions.

## Statistical analyses

To analyze each data set, we used a survival analysis with a semiparametric Cox proportional hazard model of interval censored data. In these models we had to choose one population as the reference population in the analysis. We then ran the analyses multiple times using each population as the reference population so that we could tell which populations were significantly different from other populations. In experiment 3, we had the additional variable of temperature. Here we first ran a fully factorial analysis using US1-MD and 7 °C as references to determine if significant effects of population and temperature existed. We then conducted additional tests within each temperature and also changing the reference populations and temperatures to determine which populations and temperatures were significantly different from each other. We used R version 3.2.3 (*R Development Core Team, 2015*) for all statistical analyses.

## RESULTS

### Experiment 1: the effect of population on desiccation tolerance

All of the snails from each population survived through 9 h out of water with most snails dying by 36 h (Fig. 2). We found a significant effect of population on survival (Fig. 2; Table 1). US1-WY was found to survive longer out of water than US1-PA with US1-WY surviving at a rate of about 63% with US1-PA at 28% at 18 h. US2-O29 was not found to be different in their survival rate compared to either of the other populations. No snails survived to 48 h, and 25% or less of each population survived to 24 h.

**Table 1 Results of the survival analyses comparing the survival rates of three populations of snails exposed to air.**

|  | Estimate | Exp (Est) | Std. Error | z | p |
|---|---|---|---|---|---|
| Reference population: | US1-PA |  |  |  |  |
| *vs.* US1-WY | −0.7880 | 0.4548 | 0.2715 | −2.903 | **=0.0037** |
| *vs.* US2-O29 | −0.4327 | 0.6487 | 0.2797 | −1.547 | =0.122 |
| Reference population: | US1-WY |  |  |  |  |
| *vs.* US1-PA | 0.7880 | 2.199 | 0.2835 | 2.780 | **=0.0054** |
| *vs.* US2-O29 | 0.3553 | 1.427 | 0.2436 | 1.458 | =0.145 |
| Reference population: | US2-O29 |  |  |  |  |
| *vs.* US1-PA | 0.4327 | 1.541 | 0.2900 | 1.492 | =0.136 |
| *vs.* US1-WY | −0.3553 | 0.701 | 0.2461 | −1.443 | =0.149 |

Note:
Each analysis uses a different population as the reference population. Numbers in bold indicate statistical significance.

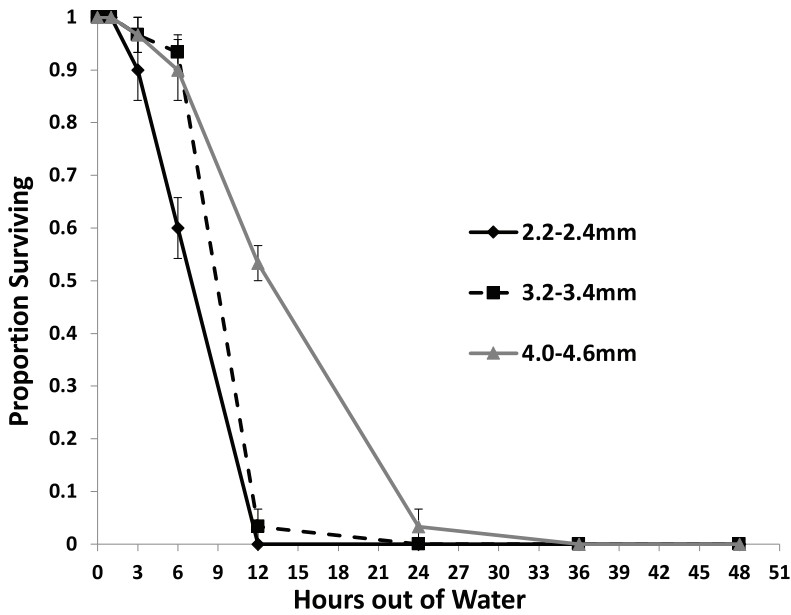

**Figure 3 The average proportion of snails surviving from the US1-PA population of three size classes over the course of 48 h out of water.** Error bars are standard errors of average proportions.

## Experiment 2: the effect of snail size on desiccation tolerance

Larger snails were found to survive significantly longer than smaller snails with no snails surviving to 36 h out of water (Fig. 3). Each size class survived at a different rate than each other size class with survival rate increasing with size (Table 2). No snails of the 2.2–2.4 mm size class survived out of water to 12 h, while over half of the 4.0–4.6 mm snails survived to that time. Only snails in the 4.0–4.6 mm size class survived to 24 h.

**Table 2 Results of the survival analyses comparing the survival rates of three size classes of snails exposed to air.**

|  | Estimate | Exp (Est) | Std. Error | z | p |
|---|---|---|---|---|---|
| Reference class: | 2.2–2.4 mm |  |  |  |  |
| *vs.* 3.2–3.4 mm | −1.174 | 0.3091 | 0.3972 | −2.955 | **=0.0031** |
| *vs.* 4.0–4.6 mm | −2.491 | 0.0828 | 0.5262 | −4.734 | **<0.0001** |
|  |  |  |  |  |  |
| Reference class: | 3.2–3.4 mm |  |  |  |  |
| *vs.* 2.2–2.4 mm | 1.174 | 3.235 | 0.4748 | 2.473 | **=0.0134** |
| *vs.* 4.0–4.6 mm | −1.317 | 0.268 | 0.3116 | −4.227 | **<0.0001** |
|  |  |  |  |  |  |
| Reference class: | 4.0–4.6 mm |  |  |  |  |
| *vs.* 2.2–2.4 mm | 2.491 | 12.070 | 0.7213 | 3.454 | **=0.0005** |
| *vs.* 3.2–3.4 mm | 1.317 | 3.732 | 0.4247 | 3.101 | **=0.0019** |

Note:
Each analysis uses a different size class as the reference population. Numbers in bold indicate statistical significance.

## Experiment 3: the effect of population on tolerance of moist conditions at different temperatures

Nearly all snails survived 60 days in moist conditions at 7 °C, and most snails survived to 60 days at 17 °C (Figs. 4A and 4B). All snails were dead by 30 days at 27 °C (Fig. 4C). There was some extraneous mortality in the experiment as some of the paper towel packets developed fungal growth. In nearly all of these cases, all of the snails died regardless of treatment or time since wrapping. The fungal growth was mostly found in the 27 °C treatment (14 out of 81 packets) but there was some in the 17 °C treatment (6 out of 84 packets) and the 7 °C treatment (2 out of 84 packets) as well. Data from these replicates were not used in the analysis.

The fully factorial model showed statistically significant effects of both population and temperature on survival with snails surviving longer at lower temperatures (Table 3). An analysis examining the main effect of temperature found that there were significant effects of temperature for all pairwise comparisons (7 °C *vs.* 17 °C, 7 °C *vs.* 27 °C, and 17 °C *vs.* 27 °C) ($p < 0.009$ in all cases), suggesting that survival rate decreased with increasing temperature. Comparing populations within temperatures found significant differences between several populations (Table 4). At 7 °C the US1-NJ was significantly different than all other populations as it was the only population not to experience any mortality at that temperature during the experiment (Fig. 4A). However, all populations experienced low mortality at 7 °C, and at 60 days all snails from all populations were found to survive. At 17 °C, there was also relatively low mortality in all populations until about 50 days (Fig. 4B). US1-MD, which experienced no mortality at 17 °C, was found to be statistically significantly more likely to survive than US1-PA and US2-Syr (Table 4). Comparisons between US2-Syr (which experienced the most mortality (about 35% at 60 days)) and US1-PA as well as between US1-NJ and US1-MD fell just short of statistical significance (Table 4). At 27 °C, there were again some differences in mortality rates. US1-

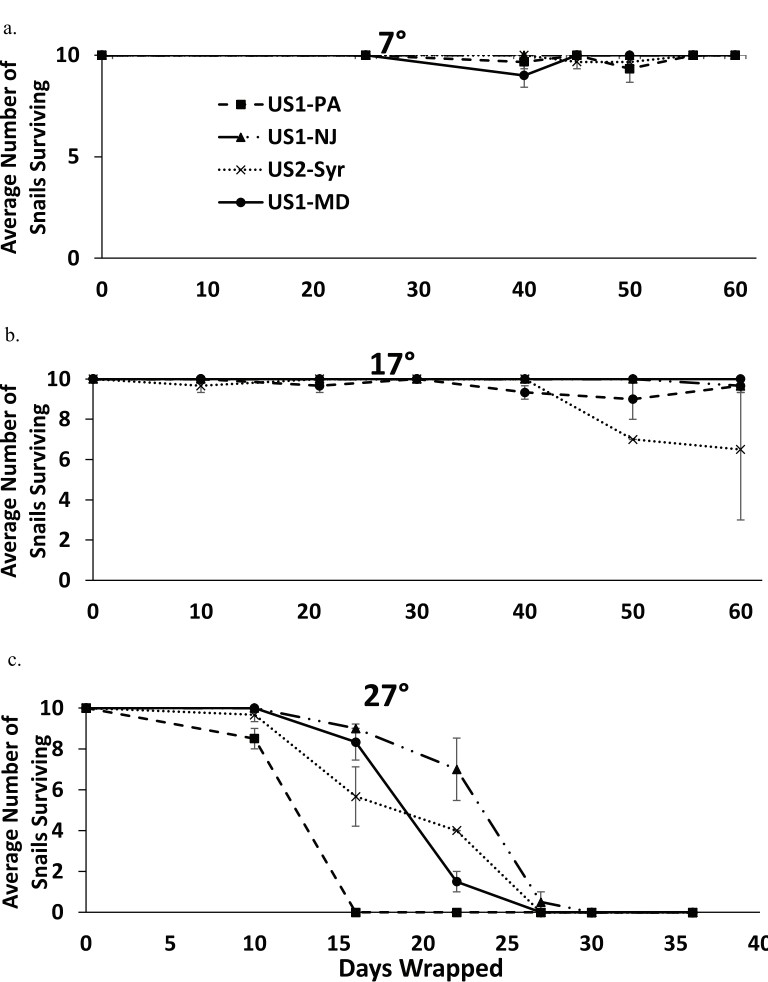

**Figure 4 The average number of snails surviving out of 10 from each population when exposed to moist conditions over time at 7 °C (A), 17 °C (B), and 27 °C (C).** Note that the horizontal scale is different in (C). Error bars are standard errors.               

**Table 3 Results of the survival analyses comparing the survival rates of four populations of snails removed from water but exposed to moist conditions at different temperatures over time.**

| Reference population: | US1-MD | | | | |
|---|---|---|---|---|---|
| Reference temperature | 7 °C | | | | |
| | Estimate | Exp (Est) | Std. Error | z | p |
| *vs*. US1-NJ | −9.845 | $5.3 \times 10^{-5}$ | 2.669 | −3.688 | **=0.0002** |
| *vs*. US1-PA | 0.1925 | 1.212 | 3.437 | 0.056 | =0.96 |
| *vs*. US2-Syr | −0.4051 | 0.669 | 4.096 | −0.099 | =0.92 |
| | | | | | |
| *vs*. 17 °C | −9.2530 | $9.6 \times 10^{-5}$ | 2.710 | −3.415 | **=0.0006** |
| *vs*. 27 °C | 8.4010 | $4.5 \times 10^{-3}$ | 3.798 | 2.212 | **=0.0269** |

Note:
US1-MD is the reference population and 7 °C is the reference temperature. Numbers in bold indicate statistical significance.

**Table 4 p-values from results of pairwise comparisons using survival analyses between populations within temperatures.**

|  | US1-NJ | US1-PA | US2-Syr |
|---|---|---|---|
| 7 °C |  |  |  |
| US1-MD | **<0.001** | =0.961 | =0.918 |
| US1-NJ |  | **<0.001** | **=0.007** |
| US1-PA |  |  | =0.889 |
|  |  |  |  |
| 17 °C |  |  |  |
| US1-MD | =0.064 | **<0.001** | **<0.001** |
| US1-NJ |  | =0.633 | =0.448 |
| US1-PA |  |  | =0.057 |
|  |  |  |  |
| 27 °C |  |  |  |
| US1-MD | **<0.001** | =0.436 | =0.908 |
| US1-NJ |  | =0.216 | **<0.001** |
| US1-PA |  |  | =0.418 |

Note:
Numbers in bold indicate statistical significance.

NJ, which appeared to have the highest survival rate (Fig. 4C), was significantly different from US1-MD and US2-Syr (Table 4). Snails from the US1-PA population did not survive to 16 days, where all other populations had at least 56% survival to that point. Even though US1-PA appeared to have the lowest survival rate, it was not significantly different from any other population at 27 °C.

## DISCUSSION

The results from these experiments show that there is variation between and within populations of NZMS in their tolerance to desiccation. Desiccation tolerance appears to be important in the invasion success of numerous aquatic gastropods (*Havel et al., 2014*). Tolerance to drying can help to facilitate overland transport *via* vectors such as birds, boats, contaminated fishing gear, *etc*. Research on desiccation tolerance in aquatic gastropods have revealed substantial variation in ability to survive out of water ranging from hours to months (see *Havel et al., 2014* for a review). In these studies, desiccation tolerance is influenced by temperature, relative humidity, the size of the snail, the presence or absence of an operculum, and other conditions (*Havel et al., 2014*). Typically, survival increases with decreasing temperature, increasing humidity, the presence of an operculum, and increasing snail size.

The results presented here demonstrate that the NZMS is very tolerant to being removed from water which has likely played a major role in its invasion success (*Proctor et al., 2007*; *Alonso & Castro-Díez, 2008*). The snail possesses an operculum that tightly fits over the aperture thus creating a seal that can help resist water loss. This trait has been credited previously for the ability of the snail to tolerate desiccation (*Wood et al., 2011*). The results also show that variation exists between populations in tolerance to desiccation
and to being out of water in moist conditions. This variation exists both between and within MLGs as different populations of the US1 genotype showed different tolerances. The fact that the snails have been kept in a laboratory in essentially a common garden for multiple generations suggests that the variation is likely genetic. Behaviors associated with predator avoidance such as geotaxis, photokinesis, and dispersal have also been shown to vary between MLGs and between populations within genotypes in this species (*Levri & Clark, 2015*; *Levri et al., 2017*; *Levri, Luft & Li, 2019*). While some studies have found that limited genetic variation exists within NZMS in their invaded range (*Stadler et al., 2005*; *Dybdahl & Drown, 2011*; *Hamada et al., 2013*), it seems apparent that this species is capable of demonstrating significant variation in traits relating to invasion success in its introduced range. The variation between populations could be the result of several processes including mutation accumulation, the expression of heat shock protein (HSP) genes (*Mizrahi et al., 2010*, *2014*), variation in shell permeability to water (*Van Aardt & Steytler, 2007*), or epigenetic effects (*Thorson et al., 2017*).

The first experiment showed that there was significant variation between populations in desiccation tolerance in a dry environment. The US1-WY population appeared to survive the longest out of water and was significantly different from the US1-PA population. This result is consistent with the fact that the US1-WY population comes from the most arid environment of the three populations. None of the snails from any of the populations survived to 48 h, and only a few snails survived from the US2-O29 population to 36 h. These results are similar to the results from previous studies. *Winterbourn (1970)* experimenting with the species in New Zealand found 100% mortality at 20–22 °C in very low relative humidity at 30 h. *Richards, O'Connell & Shinn (2004)* found 99% mortality for snails from 4 to 5 mm in length at 21 °C at a relative humidity of 20–25% at about 44 h, and *Alonso & Castro-Díez (2012)* found 99% mortality of mean length snails of 3.8 mm kept at 15 °C at about 43 h. *Collas et al. (2014)* found that *P. antipodarum* had 99% mortality at 43.5 h at 20 °C at an average relative humidity of 68.3%. *Romero-Blanco & Alonso (2019)* examined the effect of air exposure and the effect of the temperature and conductivity of the water that the snails were reintroduced to. They found that snails exposed to 20 h of drying were significantly more likely to die than snails not exposed to desiccation, but snails exposed to 20 h of drying showed no differences in mortality after 50 days of exposure to different temperatures and conductivities. However, some behavioral changes (reaction times) were noted between treatments. They also found no significant differences in neonate production between treatments.

We found that when snails were removed from water but kept moist, mortality increased over time at a greater rate as the temperature was increased from 7 °C to 27 °C. This is perhaps not surprising given that the metabolic rates of the animals increase with increasing temperature. The fact that there was almost no mortality at 7 °C for any population is consistent with the fact that these snails typically respond to cold temperatures by retreating into their shells and may overwinter this way (*Moffitt & James, 2012*). (Although *Verhaegen, von Jungmeister & Haase (2021)* demonstrated that *P. antipodarum* can be active at 2 °C). Thus, if snails are attached to items that are removed

from water and stored in a cold area (especially if encased in mud), it is quite possible that they may be able to survive for several months.

In experiment 3, we made an error in our experimental design by putting all of the packets of snails in each treatment into one box. Thus, there was one box of snails in each temperature treatment. This resulted in pseudoreplication of temperature treatment which limits the conclusions we can make about the differences between the temperature treatments because the variance within the temperature treatment is not accounted for. It is possible that the differences between the temperature treatments could be due to box effects, thus our interpretation of these results should be considered with that in mind. However, the results were consistent with expectation as the colder treatments resulted in greater survival, and the results were similar to a preliminary experiment we conducted. We found significant differences between some of the populations in tolerance to desiccation at all three temperatures with the US1-NJ population appearing to tolerate desiccation in a moist environment better than the other populations. The US1-PA population appeared to be least tolerant of desiccation at 27 °C, however, statistically it was not different from any of the other populations. This may have been due to the fact that this population had the highest rate of loss of paper towel packets due to fungal growth (five out of 21 lost), thus US1-PA had the smallest sample size (US1-NJ had three lost, US1-MD 2 lost, and US2-Syr had four lost, but two of the ones lost from US2-Syr was after 27 days when nearly all snails were dead in each treatment anyway). While it makes sense that fungal growth would have a greater effect on the warmer treatments, since the temperatures were not properly replicated, we cannot be sure if the effect of the fungus was affected by temperature.

*Winterbourn (1970)* also examined the tolerance of *P. antipodarum* to desiccation in a moist environment created in a closed petri dish with a filter paper saturated with water. He found 100% mortality by 52 days with 90% mortality by about 30 days. Survival rates were higher in the present experiment. This could be due to the differences in how the experiments were conducted. *Winterbourn (1970)* had the snails on top of wet filter paper, while the present study had the snails wrapped in a moist paper towel. The paper towel may have provided a moist environment longer than in the previous study. Another possibility is that evolution of increased tolerance may have occurred in the lineages in the present study. *Winterbourn (1970)* used snails in New Zealand. The snails used here were invasive populations that likely had to go through time out of water in order to establish those populations possibly resulting in selection for desiccation tolerance. *Alonso, Valle-Torres & Castro-Díez (2016)* attempted to simulate potential natural conditions when they removed *P. antipodarum* from water in leaf litter, artificial sediment, and clay. They found that the snails survived less than 3 days in the leaf litter, but there was greater than 20% survival after 4 days in both sediment and clay demonstrating that the material that the snails are transported in can influence the survival rate.

In addition, the results of experiment 2 are consistent with *Richards, O'Connell & Shinn (2004)* who found that larger *P. antipodarum* resist desiccation longer than smaller individuals. *Richards, O'Connell & Shinn (2004)* found that at 21 °C snails between 2.00 and 2.49 mm had 99% mortality in about 11.48 h which is very similar to our results.

However, *Richards, O'Connell & Shinn (2004)* found snails at 21 °C ranging in size between 3.00 and 3.49 mm in length had 99% mortality at 32.2 h and snails ranging in size between 4.00 and 4.49 had 99% mortality at 43.9 h. This differed significantly from our experiment where at 18 °C snails 3.2 to 3.4 mm had near complete mortality at 12 h and snails 4.0 to 4.6 mm had near complete mortality at 24 h. These differences could be due to the small differences in size ranges used in the two experiments, the slight differences in temperature, the slight differences in relative humidities (20–25% in *Richards, O'Connell & Shinn (2004)* and 19–23% in the present study), or differences between the two populations used in the two experiments.

The results suggest that to control the spread of the snail, individuals should understand that the snails can live out of water for at least several hours, and if kept moist and cool, they could live for months. While some populations exhibit greater desiccation tolerance than others, all populations can survive for a time that allows for effective dispersal whether by animal (mammals and birds) or humans if equipment, gear, and clothing are not sufficiently decontaminated. Water users should avoid wearing or using the same gear in multiple locations without efforts to decontaminate them when they are in an area known to be infested by NZMS.

## CONCLUSIONS

These experiments show evidence of variation in desiccation tolerance between populations of an aquatic invader suggesting that this trait is malleable in invasive NZMS populations. The fact that there are differences between populations of the same multi-locus genotype suggests that characterizing the traits of an invader should be done by examining populations from multiple locations, and populations from different locations should not be assumed to have the same traits including tolerances to biotic and abiotic stressors.

## ACKNOWLEDGEMENTS

We thank Jingyi Xu, Tess Woods, Maureen Levri, James Levri, Joseph Levri, Matthew Levri, and Adrianna Levri for assistance in the lab, and Mark Oswalt for logistical support. We thank Xiaosong Li for assistance with the statistical analysis. We also thank Maureen Levri and anonymous reviewers, for comments on earlier versions of the manuscript.

### Funding

This work was supported by a grant from the Mid Atlantic Panel on Aquatic Invasive Species to Edward Levri as well as grants from Penn State Altoona. The funders had no role in study design, data collection and analysis, decision to publish, or preparation of the manuscript.

## Grant Disclosures

The following grant information was disclosed by the authors:
Mid Atlantic Panel on Aquatic Invasive Species to Edward Levri.
Penn State Altoona.

## Competing Interests

The authors declare that they have no competing interests.

## Author Contributions

- Edward P. Levri conceived and designed the experiments, analyzed the data, prepared figures and/or tables, authored or reviewed drafts of the article, and approved the final draft.
- Sheila Hutchinson conceived and designed the experiments, performed the experiments, prepared figures and/or tables, authored or reviewed drafts of the article, and approved the final draft.
- Rebecca Luft conceived and designed the experiments, performed the experiments, prepared figures and/or tables, authored or reviewed drafts of the article, and approved the final draft.
- Colin Berkheimer conceived and designed the experiments, performed the experiments, prepared figures and/or tables, authored or reviewed drafts of the article, and approved the final draft.
- Kellie Wilson conceived and designed the experiments, performed the experiments, prepared figures and/or tables, authored or reviewed drafts of the article, and approved the final draft.

## Field Study Permissions

The following information was supplied relating to field study approvals (*i.e.*, approving body and any reference numbers):

Field samples were collected using a Pennsylvania Scientific Collector's permit (2015-01-0359).

## Data Availability

The raw data for experiments 1, 2, and 3 on under different tabs in an excel file.

## Supplemental Information

Supplemental information for this article can be found online at http://dx.doi.org/10.7717/peerj.15732#supplemental-information.

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
