# Peer review of "Population influences desiccation tolerance in an invasive aquatic snail, Potamopyrgus antipodarum (Tateidae, Mollusca)"

_PeerJ, doi:10.7717/peerj.15732_

## Round 0.1 · original submission · Major Revisions

Three recognized experts have assessed your manuscript and identified a number of issues that should be resolved. The most important aspect is the problem of pseudo-replication in experiment 3, but also similar in experiment 1. In any case, it is necessary to critically discuss the limitations of the chosen design, which certainly also influence the conclusions drawn from the experiments.

You should note that one of the references (line 62: Naser & Son, 2009) is potentially problematic: The shell figure in this publication (Fig. 2) looks manipulated because the identical shell is presented 12 times in this figure, slightly rotated and with partly different "damages". You might consider removing this citation.

I hope that the supportive and constructive comments of the reviewers will allow you to carry out a substantial revision of the manuscript, which is a precondition for acceptance of the manuscript.

Reviewer 1 ·

Basic reporting

Maybe some of the Tables can be summarized with the main results, and Tables with the main statistical results could be placed in suplementary material (See additional comments for details)

Experimental design

The selection of different populations in the different experiments should be justified in the manuscript (see additional comments for details).

Validity of the findings

no comment

Additional comments

REVIEWER REPORT
Population influences desiccation tolerance in an invasive aquatic snail, Potamopyrgus antipodarum (Tateidae, Mollusca)
#82657

Have you checked the authors field study permits?
Are the field study permits appropriate?
The field study permits looks appropriate. However, I don`t know the country laws to this kind of permits.

Introduction:
Line 54: remove “- -“
Line 72: Try to avoid “unpublished data”. Is there any published source on Easter US colonization by US1 MLG?
Line 93-94: Please include some data on the tolerance to air (maybe lethal time or similar parameters to understand the high tolerance of a snail without lungs. Latter characteristic is also important to note in the introduction)
Line 96: Include “animal size” to avoid misunderstanding with “population size”
Lines 97-102: Please, change “experiments” by the aims of the study. In M&M describes the different experiments according to the aims of the study.
Line 103-104: Please, specify the population/MLGs tolerance according to weather conditions of sampling sites.
Please, provide more information on the autoecology of the NZMS, for instance the respiration mechanism (no lungs), adult size and environmental tolerance to different factors. It can help to understand the species that is being studied.

Materials & Methods:
Line 109: Snail populations?
Line 110-119: A map with the original positions will help to the reader. In this map a table (or similar) could be include with the year of the population and other characteristics. The codification of populations are complex and it doesn’t help to understand and follow the study and experiments. Maybe they can be simplified, for instance the name of the State can be used for each population (in the case of New York could be NY1 and NY2)
Line 127: Please include more information on Amquel and seawater for maintenance
Line 130: Include the time that animals were kept in the one-litre plastic bins before experiments. Please, also provide information on the snail cultures. Cultures are from the same laboratory? Or they have been reared in different laboratories and transport to the laboratory where the experiments were conducted? Please, clarify this part.
Line 132-134: Please, why these populations were used? Why not the six populations?
Line 150-151: If mobility was determined, I think that the endpoint of the study is that, as an animal can be alive without mobility capacity. Please, clarify this point.
Line 153-154: Justify the use of this population in this experiment.
Line 165-167: Please, justify the use of these populations. Is it an ecological aspect? Practical one?

Results:
Maybe a picture or similar with a summary of results could be interesting. In this case, a summary of several tables can be done in only one or two Tables o Figures. The Tables with statistical results in detail can be moved to the supplementary material. In any case,Table captions for the latter have to include the populations used for the showed results.

Maybe a summary table with the main information on the survival-tolerance:
Population X small<medium=Large snail
Population Y Temperature 7ºC…….main results
Population Z Temperature 17ºC……..main results etc.
Population A Temperature 27ºC……..main results etc.

Discussion:
Line 305-307: Please, include reference/s for this part.
Line 311-314: Please, try to not repeat results, please summarize this result without cited figures or tables.
Please, unify the format of cites, as there are articles with three authors that are cited as “et al.” and another with three authors that are cited with the family name of the three authors.
Any suggestions for managers of aquatic ecosystems? Could be worse or higher risk animals from populations with the highest dessication tolerance?
Are the highest tolerance populations the most abundant populations in US? Please further discuss these points.

References:
Line 374: “Collado”
Reference of Collado 2014 with initial and final pages?
Reference Collas et al with the journal name in italic?
Line 409-411: full name of the journal?
Line 427: full name of the journal?
Line 431-432: full name of the journal?
Line 447: full name of the journal?
Line 538: Family name

Supplementary material:
Data are appropriate

Reviewer 2 ·

Basic reporting

This manuscript is well written and well presented. Introduction and Discussion place question and findings well into the context of existing information. All formal requirements seem to be fulfilled. Figures and Tables are clear and raw data have been provided.
See "additional comments"

Experimental design

The manuscript is definitely within the scope of the journal. Research question, relevance and methods have been clearly outlined and the experiments have been conducted at a high standard.
See "additional comments"

Validity of the findings

Standards of PeerJ are met.
See "additional comments"

Additional comments

I found no major problems and have only few suggestions and questions requiring some additional information (in order of appearance in ms)

Line 30: what about birds as vectors?

Lines 69ff and Discussion 273ff: Donne et al. 2020, Mol Ecol 29: 3446-3465 provide a more detailed picture of genetic variation/mutation accumulation among invasive populations based on 48 SNPs.

Lines 126ff: “Before each experiment, we maintained all populations in one-liter plastic bins in water treated with Amquel® and containing 3 ppt seawater, since these snails have been found to grow faster in 3 ppt seawater than in freshwater (Drown, Levri, & Dybdahl, 2011).”
1) 5 ppt is not unambiguous; the t could stand for thousand as well as trillion; why not 0.03% seawater?
2) It is not quite clear to me how exactly you set up the water: this was tap water (composition known?) treated with Amquel and then you added sea salt up to 0.03%? Maybe you can rephrase this sentence.
3) Drown et al. tested their snails at 0, 5, 10 etc. ppt, 3 ppt has not been in their setup.

Line 151f: What was the rational to check survival after 24 hours and after 1 week again? Snails that eventually died during that week may have died for reasons different from desiccation. You do not present statistical analyses of the 24 hours data. Any differences? Why then state that at all? Still, I would ask you to explain the waiting time of one week.

Line 236f: “However, all populations experienced low mortality at 7 °C, and at 60 days all snails from all populations were found to survive.” – This is contradictory, if you had mortality, not all would have survived for 60 days, when you terminated the experiment.

277ff: What about differential algal growth on the shells as source of intrapopulation variation?

Raw data: Maybe add a title page to the Excel file so that it can be attributed to its “origin” independently of the paper.

Reviewer 3 ·

Basic reporting

The manuscript meets the standards of basic reporting outlined by PeerJ.

Experimental design

In my review I detail a couple of issues related to the experimental design of Experiment, notably, issues related to pseudoreplication in Experiment 3.

Validity of the findings

The validity of some of the findings stemming from Experiment 3 are questionable without more explicit attention given to their shortcomings.

Additional comments

I ask the authors to consider removing Experiment 3 from the manuscript since the results of Experiments 1 and 2 constitute a nice contribution to the literature.

Annotated reviews are not available for download in order to protect the identity of reviewers who chose to remain anonymous.

---

## Round 0.2 · Minor Revisions

Thank you very much for the thorough revision which addresses the concerns of the three reviewers and my own. There are still several smaller changes to be down (cf. "Additional comments" of reviewers 1 and 2). I am happy with your discussion and the highlighting of the problem of pseudoreplication so that I vote for leaving the experiment 3 in.

I look forward to receive you revised manuscript.

Reviewer 1 ·

Basic reporting

I think the authors have satisfactorily incorporated most of the comments made in the previous review. As other reviews have noted under methodological point of view the pseudoreplication of the experiments could be a statistical problem. However, I believe that the results show a relevant utility from a biological point of view, since it shows the difference in response to desiccation of different populations of New Zealand Mud snail. This is relevant from a species dispersal point of view, since this mechanism is very relevant in the dispersion at a regional scale.

Experimental design

Pseudoreplication could be problematic from a statistical point of view, but the results are very relevant from a biological point of view. In general there are several populations that have been tested under laboratory controlled experiments, which which guarantees reliable results.

Validity of the findings

The results are novel, since the tolerance to desiccation of different invasive populations of New Zealand Mud Snail is analyzed for the first time. The results are novel, since the tolerance to desiccation of different invasive populations of New Zealand Mud Snail is analyzed for the first time. Perhaps the statistical design could be better designed. In any case, the results are important from the ecological and biological point of view.

Additional comments

Minor changes:

Abstract: include some quantitative results of the study
Line 53: popultions instead of population (s)
Line 56: remove --
Line 145-154: include the name of river and state in brackets after the site code and remove lines 148-154
Line 160 "in our lab" and remove (Levri, unpublished data)
Line 276-279: For instance, the range of survival in days at different range of temperatures could be included in the abstract as a quatitative result.
Line 390 "the tolerance of P. antipodarum to desiccation..."
Line 401: "Valle-Torres"
Line 430-433. Remove these sentences and focus on "These experiments..."
Line 517-520 Is the volume of the publication available?
Line 593-597; Include Aquatic Nuisance Species Task Force | U.S. Fish & Wildlife... and the link
Line 635. Remove the "box" in this line (is it a space?)

Reviewer 2 ·

Basic reporting

In general, the manuscript has been considerably improved through the provision of a lot of additional information requested by all referees.

Experimental design

I have overlooked the pseudoreplication issue of experiment 3 in my first review (maybe a consequence of the requested short review time?). I do agree that this is a problem and I would rather opt to leave this experiment out as we cannot know whether the boxes had an effect - in particular as there was also a fungus issue in one of them. On the other hand, I do not see that experiment one could have really suffered from a pseudoreplication problem by placing each treatment into a separate tray considering the overall arrangement the authors describe.

Validity of the findings

No further comment (but see 2. Experimental design)

Additional comments

119: Prosobranchia is a paraphyletic taxon and should not be used; instead: Caenogastropoda
176: Instant Ocean - add sea salt
421ff: Could the difference between your and Richards et al.'s (2004) findings be due to genetic differences between clones? From the geography I would assume that clonal lineages differed among studies.

Dispersal by birds: there is quite some literature providing evidence that snails can "fly" including survival of the gut passage in birds; see e.g. the work of Casper van Leeuwen for freshwater snails. Japanese colleagues have demonstrated that for small land snails.

Verhaegen et al. 2021 (Hydrobiogia 848: 2153-2168) report active P. antipodarum at 2°C (p.2158)

Reviewer 3 ·

Basic reporting

Adequate.

Experimental design

Experiments 1 and 3 are pseudoreplicated, but I appreciate that in their revised manuscript (in the Discussion) the authors have addressed this issue.

Validity of the findings

In the instances where the experimental design has elements of pseudoreplication, the authors can not say with measurable certainty whether the effects they observe are due to their experimental treatment, or other factors. In their revision the authors have brought this issue to light, and they tell readers to interpret their results with caution.

---

## Round 0.3 · accepted · Accept

Thank you very much for addressing all the reviewers' comments in your latest revision of the manuscript. I have assessed your revision myself, and I am happy with the current version. Your manuscript is now ready for publication.